# The Future of Mine Safety: A Comprehensive Review of Anti-Collision Systems Based on Computer Vision in Underground Mines

**DOI:** 10.3390/s23094294

**Published:** 2023-04-26

**Authors:** Mohamed Imam, Karim Baïna, Youness Tabii, El Mostafa Ressami, Youssef Adlaoui, Intissar Benzakour, El hassan Abdelwahed

**Affiliations:** 1Alqualsadi (Digital Innovation on Enterprise Architectures) Research Team & IRDA (Information Retrieval and Data Analytics) Research Team, Rabat IT Center, ENSIAS, Mohammed V University, Rabat 10112, Morocco; 2MASciR (Moroccan Foundation for Advanced Science), Innovation and Research, Rabat 10112, Morocco; 3Reminex (Research & Development, Engineering and Project Delivery Arm), Managem, Casablanca 20250, Morocco; 4Faculté des Sciences Semlalia de Marrakech (FSSM), Cadi Ayyad University, Marrakech 40000, Morocco

**Keywords:** anti-collision systems, collision avoidance, pedestrian detection, underground mines, computer vision, deep learning, artificial intelligence

## Abstract

Underground mining operations present critical safety hazards due to limited visibility and blind areas, which can lead to collisions between mobile machines and vehicles or persons, causing accidents and fatalities. This paper aims to survey the existing literature on anti-collision systems based on computer vision for pedestrian detection in underground mines, categorize them based on the types of sensors used, and evaluate their effectiveness in deep underground environments. A systematic review of the literature was conducted following the PRISMA (Preferred Reporting Items for Systematic Reviews and Meta-Analyses) guidelines to identify relevant research work on anti-collision systems for underground mining. The selected studies were analyzed and categorized based on the types of sensors used and their advantages and limitations in deep underground environments. This study provides an overview of the anti-collision systems used in underground mining, including cameras and lidar sensors, and their effectiveness in detecting pedestrians in deep underground environments. Anti-collision systems based on computer vision are effective in reducing accidents and fatalities in underground mining operations. However, their performance is influenced by factors, such as lighting conditions, sensor placement, and sensor range. The findings of this study have significant implications for the mining industry and could help improve safety in underground mining operations. This review and analysis of existing anti-collision systems can guide mining companies in selecting the most suitable system for their specific needs, ultimately reducing the risk of accidents and fatalities.

## 1. Introduction

Despite the technological advancements [1,2,3] of the modern world, mining is still one of the industries with the highest rates of accidents and occupational illnesses [4,5,6,7,8]. The number of accidents caused by mining equipment has risen in recent years [9]. Many workers seem to be unable to recognize hazards or interpret and recognise risk situations in terms of actions [10,11].

However, with the growth of Industry 4.0 in recent years, there has been a significant decrease in accidents and injuries, making mining a much safer industry than before. The latest technological revolutions, known as Mining 4.0 [12,13,14], have played a crucial role in this transformation, bringing about a paradigm shift in the way mining operations are carried out. Staff and machines are all safer as a result of this evolution [15]. Moreover, in Industry 5.0 [16], cobots will be the next generation of industrial robots (autonomous and agnostic to humans) with increased sensitivity to human activities and readiness to interact with humans.

One area where technological advancements have had a significant impact is in the field of artificial intelligence (AI). Throughout the entire life cycle of mining, from exploration to reclamation and closure, artificial intelligence (AI) can be implemented in various stages, such as exploration, planning, mobile machine operations, drilling and blasting, and ore beneficiation [17,18]. The utilization of AI has significantly improved the field of machine and vehicle operation automation in recent times [17,18].

Perhaps one of the most promising applications of AI in mining is in the development of anti-collision systems based on computer vision. These systems use advanced sensors, such as cameras and LIDAR [19,20], along with sophisticated algorithms and deep learning techniques to detect and track the movement of people and objects in underground mines [21,22,23,24,25,26,27,28,29,30,31,32]. This technology can help prevent accidents and injuries by providing real-time alerts and warning signals to operators and workers [22].

Another technological advancement that has gained traction in the mining industry is the use of virtual and augmented reality (VR/AR) technologies [33,34]. VR/AR can provide immersive training experiences for workers, allowing them to practice their skills in a safe and controlled environment [35,36]. It can also be used to simulate emergency situations and prepare workers for different scenarios, improving their response time and overall safety [37]. In addition, VR/AR can be used for remote monitoring and maintenance of equipment, reducing the need for workers to physically inspect machinery and thus minimizing the risk of accidents. All these interactions between the mining industry and these areas mentioned above fall into the category of disruptive mining technologies [38] as shown in Figure 1.

### 1.1. Safety Challenges in Underground Mines

Falls, bricks, rocks, loading and shipping procedures, and electrical faults are also common causes of injuries. Additionally, engine-powered machinery has a considerable effect on accident rates [39,40], especially as the need for more complicated and sophisticated devices grows, requiring a higher degree of expertise to operate [41].

Mine construction, equipment specification, and human factors all affect the risks associated with heavy machinery [42]. Staff become trapped in rolling machines, are caught by moving machine parts, or are run over by transport vehicles in these collisions (Figure 2). Keeping workers healthy when they are around machines is a difficult task. Allowing workers to be around moving equipment to perform their jobs is a common industry procedure in many ways. In underground mines and smaller surface activity, for example, personnel and massive mobile equipment are often in close proximity.

Consequently, the goal of this study is to highlight the accidents involving machinery, so we can have a better understanding of the causes in order to enhance the security of pedestrians, machines, and mining construction, including their roads and ramp design.

### 1.2. Taxonomy of Accidents in Underground Mines

The studies examined identify multiple categories of equipment that were involved in accidents, such us haulage trucks, front-end loaders, non-powered hand tools, dumpers, conveyors, continuous miners, forklifts, long walls, dozers, LHDs, jack leg drills, and shuttle cars. Accidents caused by the machines mentioned before and other machines and materials I will discuss in the next sections are classified according to the Mine Safety and Health Administration (MSHA) [43] and many studies in Figure 3. Because some of the terms seem to be similar to or even the same as the recorded problem, I opted to create a diagram of all of the terms used in each sample rather than clustering them in Figure 3 below.

Below, we clarify these types of accidents:Electrical: caused by electric current.Explosions: involving the detonation of manufactured explosives, Airdox, or Cardox and can result in flying debris, concussive forces, fumes, or the ignition or explosion of gas or dust. This category also includes accidents involving exploding gasoline vapors, space heaters, or furnaces.Fire: including unplanned fires that are not extinguished within 10 min in underground mines or 30 min in surface mines and surface areas of underground mines.Materials: involving lifting, pulling, pushing, or shoveling material and are caused by handling the material itself. The material may be in bags or boxes, or it may be loose, such as sand, coal, rock, or timber.Machinery: caused by the action or motion of machinery or by the failure of component parts. This category includes all haulage machines, such as dozers, haul trucks, front-end loaders, load–haul–dumps, dumpers, and excavators. Accidents caused by an energized or moving unit or failure of component parts, as well as collisions between machines and objects or workers, are also included.Block Fall: caused by falling material and the result of improper blocking of equipment during repair or inspection. In this case, the accident should be classified as the type of equipment most directly responsible for the resulting accident.

For a better understanding of equipment-related accidents, reports from the Mine Safety and Health Administration (MSHA) and the Current Population Survey (CPS) starting from 1995 to 2004 were tested and analyzed [39,44,45]. Table 1 sum up Rate of fatal accidents caused by mobile and haulage equipment in some countries.

During 1995–2005, Kecojevic et al. (2007) [39,45] examined all fatalities related to equipment at underground and surface mining operations in the United States. According to their findings, haul trucks were responsible for the most fatalities, 22.3%, followed by belt conveyors 9.3%.

MSHA accident data from 1995 to 2006 were used by Kecojevic, Komljenovic et al. (2008) [39,45] to measure risk for loaders and dozers. The most common dangers for loaders, according to this report, were failures to obey proper maintenance procedures and system part failures. Failure to recognize hazardous site conditions was the most common threat for dozers.

Mitchell, Driscoll, and Harrison [46] performed a systematic study of more than one hundred deaths that happened in mines all around Australia between 1982 and 1984. They found that 34% of accidents were caused by falling objects, 29% were vehicle accidents, and being hit by a vehicle or equipment part was the most frequent cause of injuries (18%).

A study conducted by Dhillon in 2009 [47] found that fatalities at quarries in the United Kingdom have been increasing. According to this study, between 1983 and 1993, 41% of all fatalities in these quarries were caused by vehicle-related incidents, such as collisions, traveling on the edge, or rollovers.

According to a study by Ural and Demirkol (2008) [48], Turkey had one of the highest fatal accident rates in the mining industry among major mineral-producing countries in 2004, with a total of 68 fatalities. The most frequent types of fatal accidents in surface mines were blasting operations (18%), controlled haulage (16%), ground collapse (14%), and machinery (12%).

A study by Eric Stemn [49] about mines in Ghana found that the top five tasks associated with injuries in underground mining were drilling (20%), charging (10%), walking (8.3%), barring/scaling (5%), and changing/adjusting (5%), which accounted for 48.3% of all underground mining injuries. The remaining injuries were caused by component/parts (17.9%), drill rigs (12.8%), rock drills/borers (12.8%), and other earth-moving equipment (12.8%).

### 1.3. Typology of Underground Mining Machinery and Its Associated Injuries

Numerous surveys [39,41,44] show that mining equipment is the top contributor to injuries in the mining industry. Moreover, several other variables, such as work-speed, demand, and load, have an impact and can lead to injuries.

In surface mining, one study showed that out-of-control vehicles were the leading cause of equipment-related deaths [45]. Braking system failures have also been mentioned. The change to machinery and equipment without handles to suit a given function were defined as accident causes by Bonsu et al. [39,41,44].

According to the study by Eric Stemn [49], the top reasons for fatal surface mining accidents in Ghana were lack of preparation (37%), seat belt usage failure (31%), poor communication (19%), and lack of control over haul roads (13%). For injuries, collisions with other cars or pedestrians or rollovers and contact with public service poles can be considered as major reasons.

Numerous recorded accidents involve heavy transport machinery, such as haul trucks [44,50,51] and dumpers [52,53], with jack leg drills having the highest accident rate [54]. In general, mobile equipment [45] often poses a visibility issue for workers, which may be due to the size and design of the equipment. Injuries often occur while handling mining supplies, such as those used for bolting tasks, where a worker comes in direct contact with or trapped by the machinery [55]. This type of incident accounts for 54% of all deaths. “Non-powered hand tools” were reported as the safest, causing only nonfatal injuries, while the use of off-road and underground machinery was the main cause of fatalities among all causes [44].

We will give more details about types of accidents caused by haulage equipment and machines in underground mining [42,44,51,52,56,57,58].

Figure 4 shows different types of mining machinery used either in surface mining or in underground mining. However, in this study we focus on underground mining machinery because we use such machinery in mining Moroccan ores, such as cobalt, copper, silver, and gold. Dumpers and load haul dump (LHD) vehicles, which serve as the primary means of transporting material within the mine, are utilized in underground mining. So, our study concentrates on the types of accidents this machinery causes.

Table 2 shows the most frequently recorded injuries involving the following pieces of equipment in underground mines: dumper, Load Haul Dump (LHD).

Table 3 shows that dumper reversals are responsible for 21% of dumper accidents, whereas front run over and collision are responsible for 29% and 18% of dumper incidents, respectively.

Table 4 shows the frequency of injuries linked to Load–Haul–Dump (LHD) equipment for each combination of activity and mechanism.

When we analyze these statistics, it becomes clear that injuries are more common among LHD drivers and that the most common injury mechanisms are collisions, which are connected with bumpy roads. The next most prevalent action at the time of injury was gaining entrance to, and especially egress from, LHDs, with falls from the machine being a major injury mechanism.

A simplified classification of the causes of injuries is introduced in Figure 5: human mistakes, maintenance issues, and environmental factors. Generally, the authors in [42,44,51,52,56,57,58] concentrate on human error, recommending providing more formations for miners to resolve the problem. Two major problems in the repair process can be noted, which are inadequate maintenance and the malfunction of mechanical components. These papers emphasize the importance of road construction, lack of visibility, and inadequate signaling as causal factors in accidents.

This study conducts a comprehensive review of anti-collision systems for detecting pedestrians in underground mines based on computer vision and artificial intelligence. The article covers an extensive range of sensors and algorithms utilized in these systems, including deep learning techniques, cameras, and LIDAR. Moreover, various data collection methods employed in training these systems are discussed. Additionally, this study evaluates different types of anti-collision solutions, such as collision avoidance, rescue operations, mapping, and navigation, along with presenting some of the industrial solutions implemented in the mining industry. Finally, we offer unique perspectives and insights on the current state of the art in the field, addressing the challenges and opportunities for future research and development in the mining industry.

## 2. Materials and Methods

The Preferred Reporting of Items for Systematic Reviews and Meta-Analyses (PRISMA) [59] is a set of guidelines and directions for reporting systematic reviews and meta-analyses based on evidence. The original goal of PRISMA was to report on reviews that evaluate the effects of interventions, but it can also be helpful for other types of systematic reviews. Other fields of knowledge quickly adopted this technique since it incorporates the criteria of scientific methodology, such as objectivity and repeatability, into the literature review process. In 2015, the authors of the PRISMA Statement made massive modification to the PRISMA [60] to emphasize the importance of the methodological component (PRISMA-P).

The most prominent journals and databases within the interdisciplinary and engineering disciplines, such as Scopus, Google Scholar, ScienceDirect, MDPI, and Directory of Open Access Journals, were used as sources of information for this study. Throughout this research, the keywords “pedestrian”, “human”, “detection”, “anti-collision”, and “systems” were established a priori, and they were combined with “mine” and “underground” separated by the Boolean operator “AND”.

During the early stages, the primary goal of this process was to determine and choose the most optimal information (based on specified research standards). The snowballing approach [61] called for the inclusion of all literature and an extension of the publication time in the second phase. During the eligibility phase, each record was evaluated against a set of inclusion criteria to determine whether it should be included in the research: articles were required to provide data for a specific time frame and include analysis of accidents and descriptions of equipment. Any articles that failed to meet these requirements were omitted from the investigation.

Following the PRISMA diagram flow as shown in Figure 6. We tracked 2000 articles through the databases cited above. After applying the exclusion criteria (see Table 5), we eliminated all articles not published between 1998 and 2022 and all articles not written in English. Then, we kept 1868 articles. Next, we removed 165 duplicate papers. We rejected 1289 articles by analyzing titles, 247 articles by analyzing the abstracts, and 45 articles because they were not available. Finally, among the remaining 95 articles, we excluded 55 articles that concern surface mines, open pit mines, and tunnels.

## 3. Proposed Solutions for Pedestrian Detection in Underground Mines

### 3.1. The Application of Computer Vision in Underground Mining Operations

Despite improvements in the technology of sensors, the use of a many types of sensors is limited in underground mines. This is due to confined space, reduced visibility, lack of illumination, dust, high temperature, and the lack of wireless communication systems.

The use of computer vision in underground mines has many applications. These applications include mapping and navigation inside underground mines, anti-collision systems for pedestrian safety, and underground mine rescue operations (Figure 7).

### 3.2. Sensory Part

Unfortunately, environmental conditions (humidity, high temperature, pressure, dust, and lack of visibility) prevent the use of many types of sensors. Nevertheless, a multitude of devices are available that can identify people and other vehicles near mining vehicles. Several of these systems have particular restrictions. Moreover, GPS (Global Positioning System) systems are frequently employed in surface operations. Because of its reliance on satellite signals, however, this technology cannot be used in an underground environment [19,20,63]. Ultra-wide band radar (UWB) detects obstacles by sending out electromagnetic waves in the radio spectrum. Pulse radar and continuous wave (Doppler radar) radar are the two forms of radar. Among their limitations, they raise false alarms and cannot distinguish pedestrians from things [19,20,63]. The use of radio waves to identify persons and items is represented by workers wearing tags with identification codes, which are detected by the reader when they reach a danger zone, triggering an alarm that warns the vehicle operator via both an audible and visual alarms. Among the inconveniences of RFID systems, they do not offer the precise location of employees, every vehicle and person must be equipped with tags, and wall roughness causes signal attenuation in UHF [19,20,63]. Ultrasonic sensors employ the sound speed to calculate the distance to an item that reflects a sent sound wave. There are many limitations in this type of sensor that limit their use in underground mines. Temperature, humidity, and angle of incidence affect the speed of sound which in turn influence the accuracy of measured distances [19,20,63]. LIDAR produces pulses of infrared light instead and calculates the time it takes for those pulses to return after reflecting off surrounding objects. A LIDAR sensor can accurately detect the distance between all objects in a scanning area up to 50 meters. It is rapid and gives an exact depth measurement [19,20,63].

On the other hand, the parts that interest us most in this review are the solutions based on computer vision and artificial intelligence.

First, we will describe the types of cameras used to solve these problems. Cameras are employed in a variety of industries to allow the driver to navigate a blind spot region in real time through a display in the driver’s cab.

#### 3.2.1. RGB Camera

An RGB (red, green, blue) camera can be employed for multiple purposes, including surveying and mapping, stockpiling volume calculations, traffic and security surveillance, inspection, and more. An RGB camera (Figure 8a) is a sensor device that captures RGB (red, green, and blue) pictures as well as a per-pixel depth report. In low light environment it needs a spotlight LED (Figure 8b).To provide depth assessment at a large number of pixels, RGB cameras employ either active stereo or time-of-flight sensing technology. The camera must be carefully chosen, taking into account the drone’s fuel consumption. For fixed-wing drones, a tiny camera is preferred because bulky gadgets cannot be carried [19,20,63].

#### 3.2.2. Infrared Camera

A thermal camera (Figure 9) is a type of camera that uses heat-sensing technology to detect and display the temperature of objects in the area of vision. Thermal cameras work by detecting the infrared (IR) radiation emitted by objects and converting that information into an image that can be displayed on a screen. This allows thermal cameras to “see” through darkness, fog, and other forms of visual obstruction, as well as detect hot and cold spots that may not be visible to the naked eye. There are several applications for thermal cameras, including security, search and rescue, military, industrial maintenance, and building energy efficiency. They can be handled, mounted on a vehicle or aircraft, or integrated into a fixed surveillance system. Thermal cameras can be useful in a number of situations, including detecting people or animals in the dark, finding hot or cold spots in a building or electrical system, and identifying sources of heat loss or gain in a building envelope. They can also be used to detect fires, locate sources of heat of flames, and monitor the temperature of industrial equipment or processes [19,20,63].

**Figure 9 sensors-23-04294-f009:**
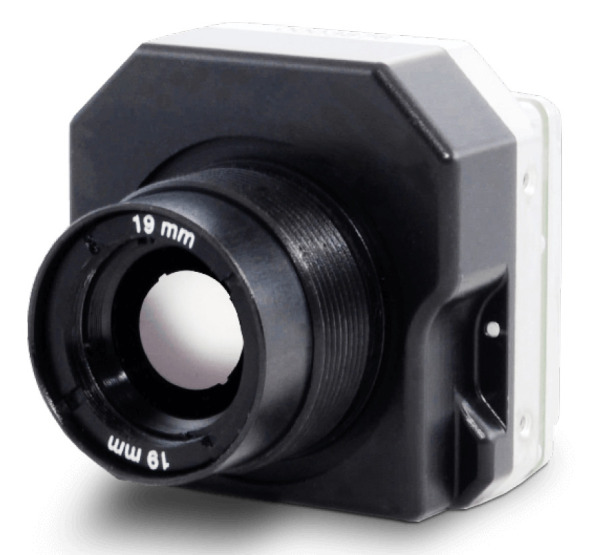
Thermal camera [66].

**Table 6 sensors-23-04294-t006:** Thermal infrared range [67].

Name	Wavelength
NIR (Near infrared)	0.75–1.4 µm
SWIR (Short wavelength infrared)	1.4–3 µm
MWIR (Medium wavelength infrared)	3–8 µm
LWIR (Long wavelength infrared)	8–15 µm
FIR (Far infrared)	15–1000 µm

As shown in Figure 10 and detailed in Table 6, the electromagnetic spectrum’s infrared (IR) band is large and has a variety of sub-bands [68], each having unique properties and applications. Commercial day/night surveillance systems and remote controllers are the most typical uses of near IR as it is simple to produce and to detect but invisible and unobtrusive. Military night vision equipment generally uses short-wave infrared radiation. Mid-wave IR is utilized for astronomy, monitoring furnaces, and photographing hot things. Long-wave IR corresponds to the wavelength of IR radiation emitted by objects that are colder than a few hundred degrees Celsius and is therefore used mostly for thermal imaging.

#### 3.2.3. Stereoscopic Camera

A camera capable of capturing three-dimensional pictures is referred to as a stereoscopic camera (Figure 11), which utilizes a combination of at least two lenses, much like the human visual system. Stereo cameras use their distinct image sensors to create three-dimensional pictures. Stereo cameras offer the benefits of high precision and high-quality resolution in a controlled environment. Under smoke, fog, or dust, however, it performs poorly because light waves are warped in these situations [19,20].

#### 3.2.4. LIDAR (Light Detection and Ranging)

Similar to radar, LIDAR (Figure 12) uses pulses of infrared light rather than radio waves to conduct remote sensing and range operations. After the pulses bounce off surrounding objects, the return time is measured. Knowing the speed of light, the LIDAR sensor can accurately calculate the distance to each object from the time between the emission of the laser pulse and the return pulse. Every second, LIDAR takes millions of precise distance measurement points from which a 3D matrix of its surroundings can be produced. This detailed mapping can provide information on the position, shape, and behavior of objects.

### 3.3. Data Collection Method

We talked about the sensory part in the previous subsection. Now, we will discuss the data collection method. Researchers have conducted various experiments using four hardware systems: aerial vehicles (drones), unmanned ground vehicles (UGVs), mobile machines, such as load–haul–dump (LHD) vehicles, and fixed cameras such as surveillance cameras.

#### 3.3.1. Drones

Drones, or unmanned aerial vehicles (UAVs) (Figure 13), can also be used in underground mines as a way to improve safety, efficiency, and productivity. These drones are typically equipped with sensors and cameras and are controlled remotely by a human operator or through autonomous programming. There are various uses for drones in underground mines, including [19,72] surveying, mapping, monitoring, inspection, hazardous tasks, and search and rescue. Globally, the use of drones in underground mines can help improve safety and efficiency by reducing the need for human workers to perform tasks in potentially hazardous environments.

The purpose of this research [23] is to present a convolutional neural network (CNN) method for low-cost micro aerial vehicle (MAV) platforms to autonomously navigate through a dark underground mine. The suggested CNN component uses the picture stream from the onboard camera to generate online heading rate directives for the MAV, allowing a collision-free descent of the tunnel axis for the platform. The production of the dataset utilized to train the CNN is a new aspect of the presented approach.

In order to deliver useful data in textureless [24], dark, and obscurant–filled conditions, the combination of optical and thermal cameras, LIDAR, and inertial measurement unit cues is examined. In northern Nevada, the integrated system is being evaluated in field experiments inside underground metal mines.

In this paper [73], during the development of the system, E. Jones et al deployed a hovermap autonomous flight system in underground and GPS-denied areas. A drone equipped with a LIDAR sensor was utilized in the development of this system. The primary goal of the examples was to increase safety by better understanding the rock mass behavior and failure mechanisms frequently observed in deep and high-stress mining situations and by incorporating these insights into the design process.

The paper [74] attempts to analyze the potential for creating a completely autonomous enclosed drone-based remote monitoring system to survey inaccessible zones in underground mines. Flying a drone underground is highly challenging because of the tight quarters, poor sight, air velocity, dense concentration of dust, absence of a wireless connection system, and other factors. Furthermore, it is very hard for a drone operator to access hazardous areas in underground mines. By suggesting the construction of an autonomous spherical micro-drone for monitoring underground mine safety, this study proposes solutions to the mentioned problems.

In this study [75], only a UAV with technology that can film with effective lighting is able to remotely visit regions at the Cuiabá mine using the sublevel technique. Additionally, the equipment must be able to fly without GPS, do so at a varied low speed to absorb minor bumps from the rock wall and other support structures, and do so often without the pilot’s visual input. Additionally, the UAV controls must be simple enough for miners with no aeronautical experience or understanding to use. For this objective, a hybrid device was created that brings together a helium-filled balloon and quadcopter propellers. It is equipped with remote control capabilities, rechargeable batteries, powerful LED lighting, image stabilization cameras, and radio frequency transmitters for both control and visualization of images. To prevent issues specific to mine operations, the gadget has been customized with a variety of additional solutions. The local team changed projects, rethought pillar sizes, discovered ore locked inside the stopes, and identified other critical circumstances as a result of the UAV’s photographs being shown in endless trails, increasing safety and output.

#### 3.3.2. Unmanned Ground Vehicle

Unmanned ground vehicles (UGVs) are becoming increasingly popular in underground mines as a way to improve safety (Figure 14), efficiency, and productivity. These vehicles are equipped with sensors and cameras and are typically controlled remotely by a human operator or through autonomous programming. Some potential uses for UGVs in underground mines include [76,77] surveying and mapping, material handling, and hazardous tasks. Overall, the use of UGVs in underground mines can help to improve safety and efficiency by reducing the need for human workers to perform tasks in potentially hazardous environments.

The authors of [22] presented many scenarios involving underground mine rescue. This paper presents the outcome of testing a UGV robotic system equipped with a sensory system and an image processing module. The module is based on an adaptation of the you only look once (YOLO) and histogram of oriented gradients (HOG) algorithms.

For belt conveyor maintenance, the authors of [78] recommend using an unmanned ground vehicle (UGV) equipped with both RGB and thermal cameras and a localization system.

In this study [25], the efficacy of a system for identifying road signs using machine vision and an autonomous vehicle for driving in underground mines was evaluated using a combination of data from cameras and LIDAR sensors.

In the field of mine safety, coal mine rescue robots with binocular vision have been tested. This work [76] comprehensively introduces the scientific state of stereo vision matching for binocular vision and camera calibration.

In this investigation [26], the accuracy of three techniques for determining the location of an autonomous driving robot in underground mines was compared. These techniques were an inertial measurement unit combined with encoder sensors (IMU + encoder), light detection and ranging with encoder sensors (LIDAR + encoder), and a combination of IMU and LIDAR with encoder sensors (IMU + LIDAR + encoder). As the robot moved, its position was calculated using each of these techniques, and the precision of the results was assessed by comparing the estimated location with the robot’s actual location.

This research [28] analyses the specifications and suggests a robot design capable of autonomous underground navigation and object manipulation. A robust four-wheeled platform with weather-resistant electric motors serves as the robot’s base. The kit comes with illumination, color and depth cameras, laser scanners, an inertial measurement unit, and a robotic arm. Two experiments were conducted to evaluate self-navigation and mapping.

Geothermal heat causes hot walls in deep gold mines, and thermal imaging becomes a viable alternative for structural imaging. The authors in [77] use the combination of temperature and 3D data to allow for the calculation of a risk measure, which may be used to identify possible danger zones. The miner’s interpretation of risk data might be represented in a variety of ways. Finally, a distance sensor and a thermal camera are utilized to detect and monitor pedestrians in order to forecast probable accidents.

In this research [79], the authors suggest PftNet, a novel parallel feature transfer network-based detector that outperforms one-stage [80,81,82,83] approaches while maintaining the accuracy of two-stage [84,85,86] methods. The pedestrian identification module and the pedestrian location module are two interrelated components that make up PftNet. The former tries to generally alter the anchor box’s location and size, filter out the negative anchor box, and improve the regression’s initialization.

The paper [87] presents an improved SLAM algorithm, called LeGO-LOAM-SM, which is designed to construct more precise point cloud maps using LIDAR sensors and improve position estimation in underground coal mines. The proposed algorithm is evaluated with experimental data, showing better map clarity, trajectory smoothness, and accuracy. Moreover, the study explores a two-dimensional occupancy grid map, which effectively filters outlier noise, and achieves high mapping accuracy of 0.01 meters while minimizing storage space requirements.

#### 3.3.3. Mobile Machines

A load–haul–dump (LHD) [88,89] is a type of underground mining vehicle that is used to load, haul, and dump material within an underground mine (Figure 15). These vehicles are designed to operate in tight spaces, such as mine tunnels and chambers. LHDs are typically used to transport materials, such as ore, waste rock, and coal, within the mine. They are equipped with a bucket or scoop on the front of the vehicle, which is used to load material into the vehicle. The material is then transported to a designated dumping location, where it is unloaded using the bucket or a conveyor belt. LHDs are commonly used in underground mines to improve efficiency and productivity. They can reduce the need for human workers to manually load and transport materials and can operate around the clock if necessary. They are also capable of operating in a variety of underground environments, including mines with low headroom or limited access.

The proposed collision detection system [62] combines a three-dimensional (3D) sensor with a thermal infrared camera for human identification and tracking. A distance sensor provides depth information in addition to a thermal camera, allowing the vehicle and pedestrian velocities to be estimated.

In a simulated environment, the control of an underground loader was analyzed using reinforcement learning and a multi-agent deep neural network approach [27]. At the initiation of each loading cycle, an agent was responsible for selecting the excavation position from a visual representation captured by a depth camera of a pile of fragmented rock. In order to fill the bucket at the prescribed loading location without colliding, getting stuck, or losing ground traction, a second agent was in charge of maintaining vehicle control.

The authors of this work [90] provide many new vision-based algorithms for visual place recognition on underground mining vehicles that outperform existing technologies while simply requiring a camera input. By selectively processing picture areas, the authors describe a Shannon entropy-based salience creation strategy that improves the performance of single-image-based location identification. The authors then include a learning-based approach that filters problematic photos from both the reference map databases and live processing using support vector machines (SVMs).

This research [91] describes a human detection strategy for preventing accidents in mines that are dark or dimly lit. This technology employs a mix of infrared sensors and distance estimation through sound or light. When there is no light, the approach is used to ensure both human recognition and precise distance measurement. The outcomes of the experiments are shown, demonstrating the strategy’s efficacy and efficiency.

For monitoring the safety and real-time location of underground miners, the enhanced YOLOv4 model reaches 96.25% and 48.2 fps in global AP (average precision) and detection speed, respectively, according to experimental results in this study [29]. The enhanced YOLOv4 model also has the benefits of excellent robustness and generalization capacity, which make it ideal for detecting individuals underground and offering a solid assurance for the management and monitoring of mine workers’ safety.

One of the most critical technologies is the detection of the obstacles in front of unmanned electric locomotives while they operate which is of great significance for safe operations. Authors of [30] proposed an improved YOLOv3 (YOLOv3-4L) algorithm for intelligent obstacle detection. The unsafe region for driving an electric locomotive can be found by locating the location of the track and extending a specified distance outside of it. Obstacles are found using the enhanced YOLOv3 algorithm, and only those types and places that coincide with risky zones are output. The experimental findings demonstrate that conventional computer vision approaches, such as perspective transformation, sliding windows, and least square cubic polynomials, can detect both straight and curved tracks, making up for the Hough transforms’ limitations in this regard. The YOLOv3-4L algorithm enhances the MAP (mean average precision) by 5.1% and the detection speed by 7 fps when compared to the original YOLOv3 method. The YOLOv3-4L detection model can match real-world working conditions and serve as a technical guide for autonomous electric locomotive operation in underground coal mines thanks to its high detection accuracy and speed.

The purpose of this study [92] is to investigate the analysis of moving targets in coal mines using computer vision processing and to explore image processing techniques for target monitoring. The paper presents the concept of image enhancement and its related algorithms, with experimental results showing that the proposed algorithm outperforms existing techniques by 56.60% and 68.26% for histogram equalization and dark primary color prior dehazing algorithms, respectively. The study also provides examples of image enhancement in coal mines and concludes that the proposed algorithm yields superior results.

In response to the need for real-time detection of obstacles in front of underground track mine carts, a new system [32] was proposed that utilizes both camera and LIDAR information. The system employs a custom point cloud clustering algorithm specifically designed for the challenging mine environment to extract obstacle information and then leverages the YOLOv5 algorithm to identify obstacles in the resulting images.

Imam et al. [93] propose a new anti-collision system for pedestrian detection in underground mines based on RGB images collected in five different mines. They used Yolov5s which reaches 75% of precision and 71% of MAP.

The paper [94] introduces a new method for solving the problem of local path planning in reactive navigation of underground vehicles. The proposed approach uses 2D LIDAR to extract the centerline of the current laneway from the skeletons obtained from the LIDAR data. The centerline is then smoothed to produce the current planned local path in underground mining scenarios.

#### 3.3.4. Surveillance Cameras

Surveillance cameras can be used in underground mines to monitor and assess the condition of equipment, infrastructure, and the environment within the mine (Figure 16). These cameras can be placed in strategic locations throughout the mine and can be used to transmit real-time video footage to a central monitoring station. There are several potential benefits to using surveillance cameras in underground mines: improved safety, enhanced productivity, and improved communication. Overall, the use of surveillance cameras in underground mines can help to improve safety and efficiency by providing mine operators with a better understanding of conditions within the mine.

The authors of [21] integrate YOLOv2 with the FCN skip structure to increase the accuracy of identifying pedestrians in groups in underground coal mines. In this approach, they offer YWSSv1 and YWSSv2, two modified variants of YOLOv2.

The problem of numerous noise sources in underground mine videos is addressed in this work [96], which proposes an improved K-SVD object identification method.

Limei CAI et al. [97] presented a method for finding miners by detecting their helmets in this research. Background subtraction, denoising, erosion, region increasing, circle detection, and tracking are among the processes. Real underground coal mine footage was used to test the method.

In this research [95], the authors present a human tracking approach for underground coal mines using scene information fusion, with the goal of improving tracking accuracy and resilience in the face of difficult elements (e.g., low illumination and low intensity contrast, bright spots, and shadow disturbance).

Zongwen Bai et al. [98] present an efficient hybrid feature detection in images approach for seamless stitching of industrial coal mine surveillance footage taken in real-world scenarios using multiple cameras. This approach utilizes valuable information from several cameras to achieve fast video stitching.

In this paper [31], an improved algorithm of mine pedestrian detection yolov4-tiny-SPP based on yolov4-tiny was proposed. This algorithm consists of adding SPP (spatial pyramid pooling) after the arknet layer. This algorithm solved the occlusion of pedestrians in mines and it reached 91.98% in MAP, which was 2.32% higher than the original algorithm.

A low-cost battery-free localization scheme called MineBL was proposed [99] for accurate positioning in underground coal mines. The approach utilizes reflective balls as position nodes and applies a series of algorithms based on depth images to achieve target localization. Moreover, an optimized weighted centroid location algorithm based on multilateral location errors was developed to minimize localization errors in underground scenarios. The experimental results demonstrate that MineBL performs well with localization errors below 30 cm in 95% of cases.

### 3.4. Algorithmic Part

As previously mentioned, we discussed the sensory systems and the data collection methods knowing we are going to quote the algorithms (Table 7) used in research already conducted to solve this problem by using computer vision in underground mines either for pedestrian detection, for navigation, or mapping anti-collision systems.

As shown in Table 7, we cite algorithms with the types, as well as the data used and the objectives of these algorithms. Additionally, Table 8 shows the AP that all these algorithms reached. Yolov4 has the highest precision in relation to the number of frames per second.

On the other hand, we know that an underground mine contains very severe environmental conditions, such as dust, dew drops, lack of visibility, and high humidity. So, the only environment that has conditions close or similar to an underground mine is road scenes at night or in severe weather conditions, such as fog and rain. We note some research on this problem (See Table 9).

The efficiency of existing pedestrian detection algorithms is severely limited by decreased visibility and fuzzy outline and look like pedestrian pictures taken during foggy weather, as described in [102]. Three innovative deep learning techniques based on Yolo are presented by the authors. Their network was made more efficient by using depthwise separable convolution and linear bottleneck techniques to minimize the computing cost and number of parameters. The authors presented a 16-bit thermal data collection collected during severe weather conditions in the four largest European Union nations in [103]. We evaluated and supplied 16-bit depth changes for the YOLOv3 DNN-based detector, with a MAP of up to 89.1%. Moreover, the experimental findings in [104] demonstrate that CNN-based detectors achieved high performance on FIR picture at night owing to the large amount and diversity of training data. The authors of [105] propose a deep-learning-based data augmentation technique that uses the six most accurate and fastest detectors (TinyV3, TinyL3, YOLOv3, YOLOv4, ResNet50, and ResNext50) to enrich far-infrared images collected in good weather conditions with distortions similar to those caused by bad weather. Multi-spectral pictures of color thermal pairs were shown to be more successful than a single color channel for pedestrian identification in [106], particularly under difficult lighting circumstances. The authors discovered that the confidence in detecting pedestrians in color or thermal pictures is proportional to lighting conditions. They propose an illumination-aware faster R-CNN with this in mind (IAF R-CNN). To offer an illumination measure of the input picture, an illumination-aware network was introduced. The study [107] uses an image feature extractor based on histograms of oriented gradients followed by a support vector machine (SVM) classifier to evaluate pedestrian recognition in low resolution night vision infrared pictures.

Instead of employing a strict intensity threshold, the authors in [108] created areas of interest by merging picture segments and exploiting infrared images’ low-frequency characteristics. It helped to make the technique less reliant on light and more adaptable.

A technique for identifying and tracking pedestrians using a single digital infrared camera was demonstrated in [109]. In this example, an assessment was made to distinguish between objects identified as hot spots and humans through the use of a support vector machine (SVM) algorithm.

Ref. [110] contains an example of a detection system’s module for pedestrian recognition, as well as some ideas for possible enhancements to evaluate human presence in both nocturnal and daytime situations.

The authors of [111] emphasized the need of integrating high-resolution imaging with range data to improve detection system performance.

The authors demonstrated a way to extract form characteristics from segmented stereo vision images in a subsequent paper devoted to a system for detecting, localizing, and tracking individuals for military UGVs [112].

An adaptive Boolean-map-based saliency model was then included to improve thermal image recognition of people [113], which allowed for better differentiating a person from the background.

In a recent paper, the authors of [114] demonstrated that an improved pedestrian identification technique based on a convolutional neural network is accurate in diverse weather situations when compared to five other standard methods. Below, we cite the research already conducted to solve this problem using computer vision.

Ref. [115] provided another example of an advanced rapid detection system that may be used to identify people (among 9000 item types).

In [116], both a thermal and an RGB camera were used for referencing infrared images.

As shown in Table 9, we cite algorithms with the types as well as data used and the objectives of these algorithms. Additionally, Table 10 shows the AP reached by all there algorithms. Yolov5s had the highest AP, and yolov4 reached the highest MAP.

According to Table 11, different solutions use an RGB camera and a thermal camera for pedestrian detection and a stereo camera and LIDAR for depth estimation. Algorithms used for pedestrian detection are Yolo and its variation (v3, v4, 9000 and tiny), Fast R-CNN, HOG, and SVM. In addition, stereo matching algorithms are used for depth calculation. In the end, the final result is pedestrian detection and pedestrian distance prediction.

**Table 9 sensors-23-04294-t009:** The most commonly used algorithms in road scenes under severe weather conditions with their purpose.

Algorithm	Data Type	Purpose
Yolo (v1, v2, v3, v4 and v5) [102,103,105,113,117]	RGB Images [102], Thermal Images [103,105,113,117]	Pedestrian detection [102,103,104,105,106,107,108,109,110,111,112,113,114,117,118], falling elderly people [118]
Tiny (v3,l3, v2) [103,105,113]	Thermal Images [103,105,113]	
Fast R-CNN [104,106]	RGB Images [106], Thermal Images [104,106]	
HOG [107,110,113]	RGB Images [110], Thermal Images [107,110,113]	
SVM [107,108,109,110,113,118]	RGB Images [110,118], Thermal Images [107,108,109,110,113]	
ROI [111,112]	Stereoscopic Images [111,112]	

**Table 10 sensors-23-04294-t010:** The most used algorithms with the accuracy they have reached.

Reference	Algorithm	AP (%)	MAP (%)	fps
[102]	Yolov3	78	N/A	22.2
	VggPrioriBoxes-Yolo	80.5	N/A	81.7
	MNPrioriBoxes-Yolo	80.5	N/A	151.9
[103]	Yolov3	N/A	80.5	
	TINYv3	N/A	66.3	
[105]	TINYv3	N/A	73.26	55.57
	TINYL3	N/A	80.14	43.01
	Yolov3	N/A	80.48	17.88
	Yolov4	N/A	86.05	15.97
	ResNet50	N/A	81	19.82
	ResNet50	N/A	77.07	17.7
[113]	Tiny Yolov2 with ABMS	87.12	N/A	62.8
	Tiny Yolov2 with BMS	85.37	N/A	62.8
	Tiny Yolov2 without preprocessing	78.4		63.8
[117]	Yolov5s	90	N/A	N/A

**Table 11 sensors-23-04294-t011:** Summary of solutions.

Purpose	Sensors	Algorithms
Pedestrian Detection	RGB Camera Thermal Camera	Yolo Fast R-CNN HOG
Depth Estimation	Stereoscopic Camera Lidar Sensor	Stereo Matching Point cloud image

### 3.5. Industrial Solutions

Over the past several years, many businesses have made investments in collision avoidance systems (CAS; known also as proximity detection systems) in an effort to decrease the frequency of accidents involving injuries and fatalities.

The Earth Moving Equipment Safety Round Table (EMESRT) is one organization that focuses on vehicle interaction and has done so for over ten years. According to the EMESRT, approximately 30–40% of mining industry deaths in underground operations each year are due to failures of vehicle interaction controls, with about half of these involving pedestrians [119].

The EMESRT realized a project that initially focused on awareness, advice, and intervention (PDS/CAS) technologies. In a nine layer model of control effectiveness (see Figure 17), the project was scattered to add both the operational controls and design. Level 9 (L9) of PDS solutions will introduce technologies that automatically intervene and take control of the machine in order to prevent or mitigate unsafe interactions.

In 2015, a report from the EMESRT highlighted some solutions for underground vehicles. The report highlighted that underground machines have struggled to achieve interoperability. In fact, the development of proximity detection systems linked to the machine’s interface has been customized for each individual machine. This creates challenges for operating in mines, as there will never be just one brand or type of equipment. Therefore, it is crucial to have interoperability between the various levels of sensing and intelligence rules and the various vehicles in order to achieve a comprehensive solution for the mining industry.

Several companies offer a wide range of advanced collision avoidance systems that can intervene in the maneuverability of machinery by braking or decreasing speed in order to prevent collisions or run-overs (LEVEL 9 Intervention Control—EMESRT) (see Figure 18). These solutions are shared and available for the international mining community in order to obtain a clearer vision on how things work.

Finally, Table 12 provides a brief summary of most CAS products applied in the mining industry available on the market. Operator fatigue can also lead to collision accidents. Table 13 summarizes some of the industrial solutions that alert operators during their work.

## 4. Discussion

### 4.1. General Discussion

Underground, mine workers usually face significant problems, such as weak lighting, heat and humidity, dirty air, and of course cramped space. All these environmental limitations prevent several types of sensors, such as GPS, RADARS, ultrasonic sensors, and RFID systems, from working in similar places. For this reason, researchers have migrated to solutions based on computer vision using all types of RGB, thermal infrared, and stereoscopic cameras as well as LIDARs. The aim of this study is presenting and explaining all these technologies and techniques used in anti-collision systems based on computer vision for underground environments and citing the proposed solutions for pedestrian detection in underground mines.

Finally, for obtaining good results for pedestrian detection in deep underground mines, the hybridation of sensors is the better solution such as the mixtures below:RGB Cameras + Thermal Cameras + Stereoscopic Cameras;Thermal Cameras + Stereoscopic Cameras;RGB Cameras + Lidar Sensor;Thermal Cameras + Lidar Sensor.

The objective of this hybridation is that each device covers the deficiency of the other device.

For the algorithmic part, Yolov3, v4, and v5 reach the highest precision and efficiency in terms of fps. In addition, they have high precision while using different types of images:RGB images;Nir images;LWIR images;Fir images.

### 4.2. Improving Mine Safety Based on Computer Vision

In this section, we propose several ideas to improve the performance and accuracy of existing collision avoidance systems using computer vision techniques. Connected mines [12,13,14,121,122] can be equipped with smart sensors that can monitor the environment and send real-time alerts to miners in case of danger. Cloud computing [123,124] can be used to store and analyze data collected by sensors, allowing anti-collision systems to benefit from additional computing power to improve accuracy and responsiveness. A multi-sensor system [22,62,125] uses a combination of sensors, such as cameras, radar, LIDAR, and proximity sensors to collect information about the environment around the mobile mining machines and allowing miners to more easily detect potential hazards. 3D mapping [87,94,125] can be used to create accurate models of the underground environment, improving obstacle and pedestrian detection. Smart robots [22,78,126] can be used to inspect dangerous or inaccessible areas, reducing the risk to miners.

The use of augmented reality [33,34,127] can help miners visualize potential obstacles and hazards, improving their real-time decision making. Finally, improved lighting conditions can help increase the accuracy of computer-vision-based anti-collision systems by providing better visibility in underground areas.

## 5. Limitations

Despite the efforts made to carry out this work, the comprehensiveness of this study cannot be guaranteed in a rather vast and evolving field.

## 6. Conclusions

### 6.1. General Conclusions

In conclusion, this review paper has analyzed the types of anti-collision systems based on computer vision for pedestrian detection in underground mines. From the results of this study, anti-collision systems based on computer vision have the potential to greatly improve safety in underground mines, particularly in pedestrian detection. The different types of sensors, such as RGB cameras, stereoscopic cameras, infrared cameras, and LIDAR sensors, used to collect data and prevent collisions between miners and mobile machines as well as their installation in UGVs, mobile machines, and camera surveillance, were thoroughly discussed. Additionally, this paper has cited some industrial solutions based on computer vision that have been implemented in underground mines to enhance safety.

### 6.2. Practical Application

Furthermore, in terms of practical applications, this paper has discussed the need for continuous innovation and development of new systems to keep miners safe and has suggested that top management’s engagement can help recognize key points that may become bottlenecks in any output rise. The biggest concern that the study should focus on is to improve the protection of machine components and proximity prevention systems to avoid collisions between machines and miners or between machines and machines.

### 6.3. Current and Future Trend

As the world moves towards Industry 4.0 and 5.0 with greater emphasis on human–machine interaction, it is important to continue to develop and innovate new systems, methods, and solutions for anti-collision systems in underground mining. The concept of autonomous mining is spreading quickly in various applications, and future research should examine how anti-collision systems can be integrated with autonomous mining to minimize accidents and achieve a digital mine that prioritizes safety.

## Figures and Tables

**Figure 1 sensors-23-04294-f001:**
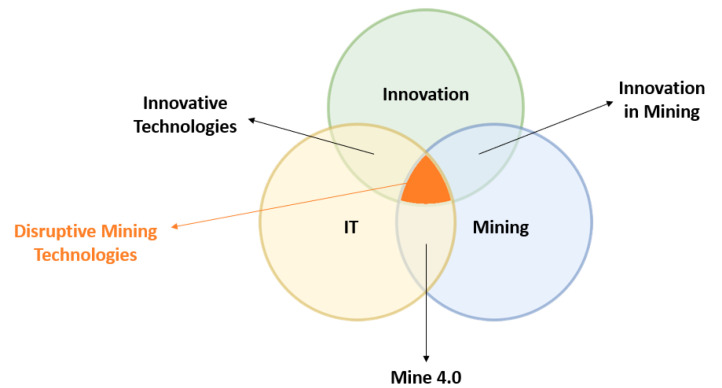
Disruptive mining technologies.

**Figure 2 sensors-23-04294-f002:**
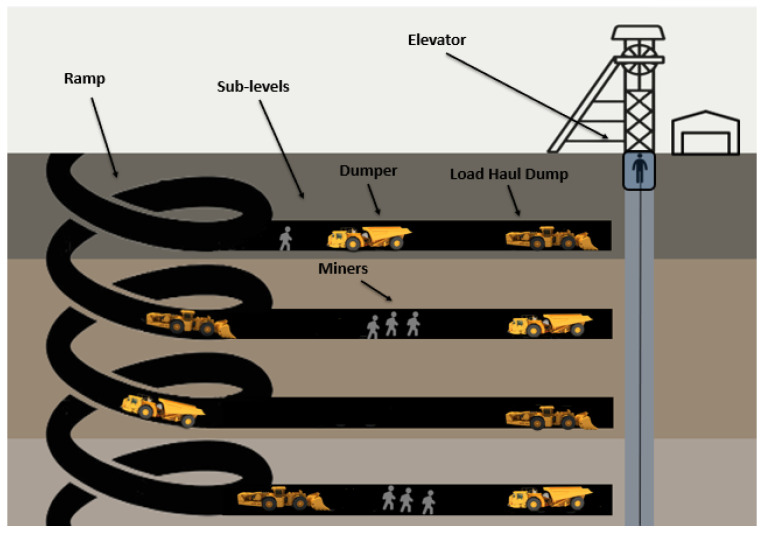
Underground mine description.

**Figure 3 sensors-23-04294-f003:**
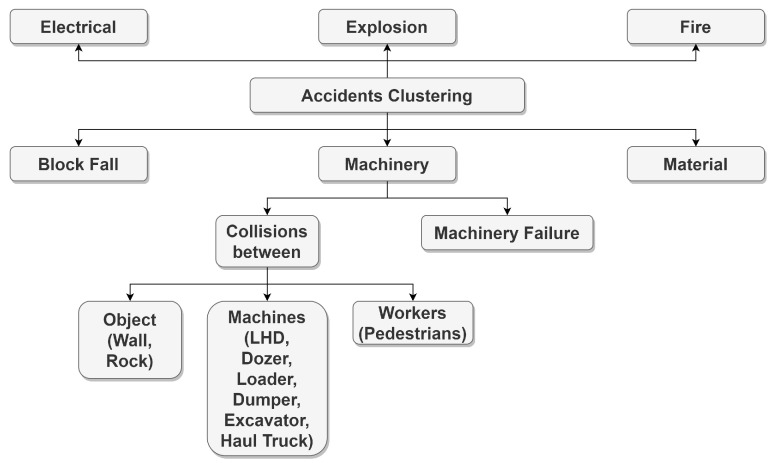
Accidents clustering.

**Figure 4 sensors-23-04294-f004:**
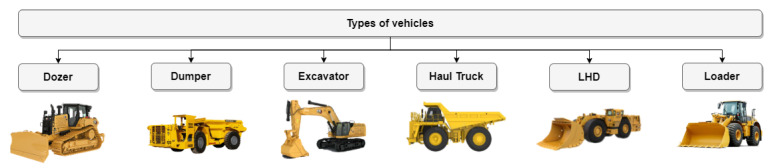
Types of equipment.

**Figure 5 sensors-23-04294-f005:**
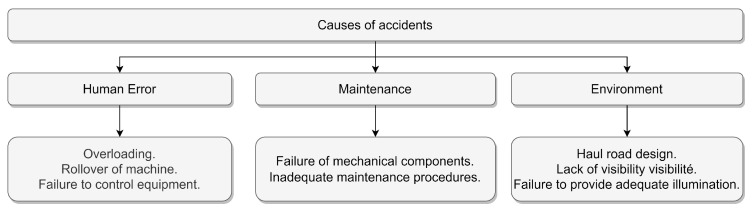
Causes of accidents.

**Figure 6 sensors-23-04294-f006:**
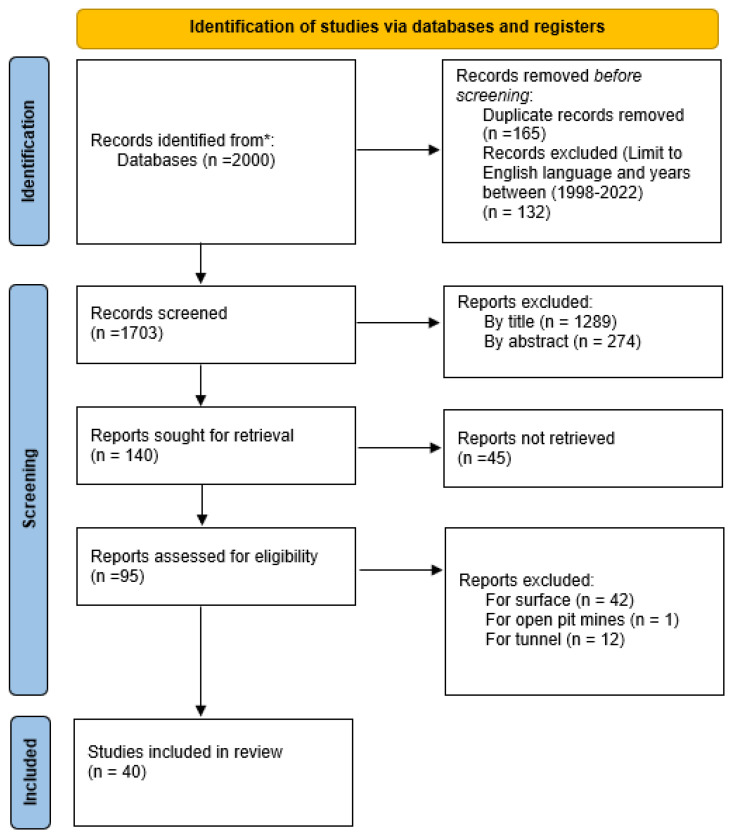
PRISMA flow diagram.

**Figure 7 sensors-23-04294-f007:**
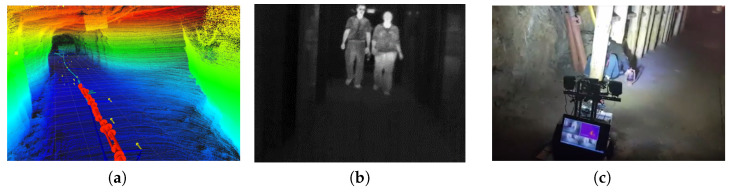
Applications of computer vision in underground mining operations: (**a**) navigation mapping [24]; (**b**) pedestrian detection [62]; (**c**) rescue operation [22].

**Figure 8 sensors-23-04294-f008:**
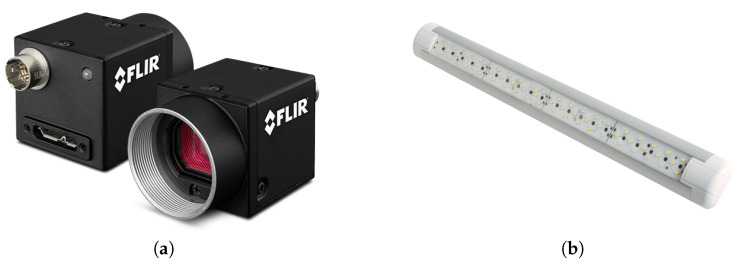
The use of RGB camera with Spotlight LED. (**a**) RGB camera [64]; (**b**) spotlight LED [65].

**Figure 10 sensors-23-04294-f010:**
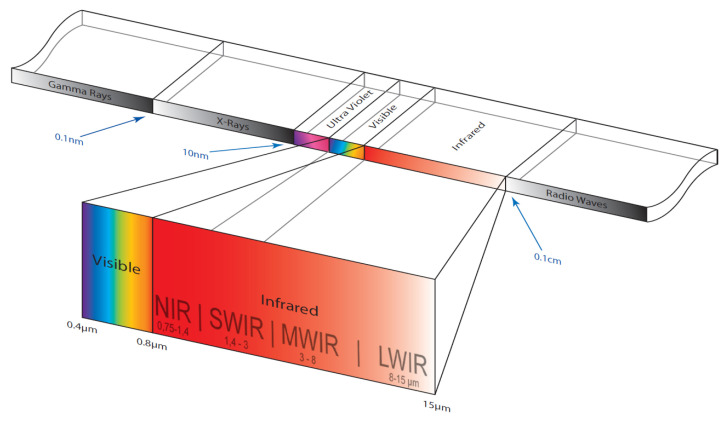
Thermal range [67].

**Figure 11 sensors-23-04294-f011:**
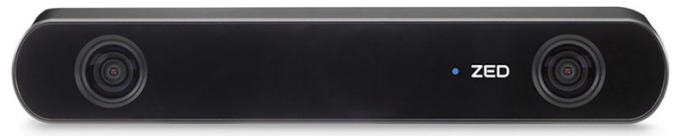
Stereoscopic camera [69].

**Figure 12 sensors-23-04294-f012:**
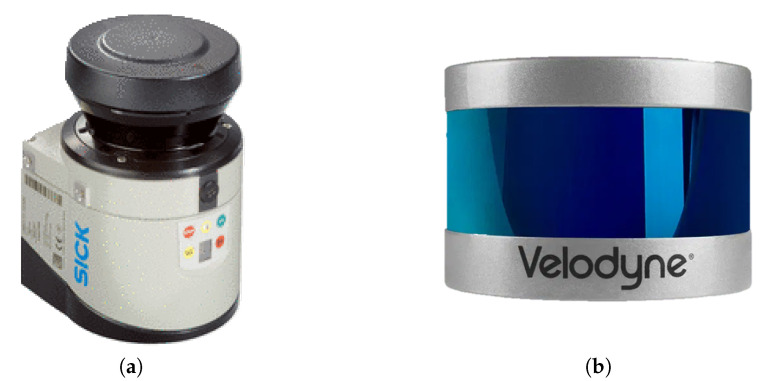
Three simple graphs. (**a**) 2D LIDAR sensor [70]; (**b**) 3D LIDAR sensor [71].

**Figure 13 sensors-23-04294-f013:**
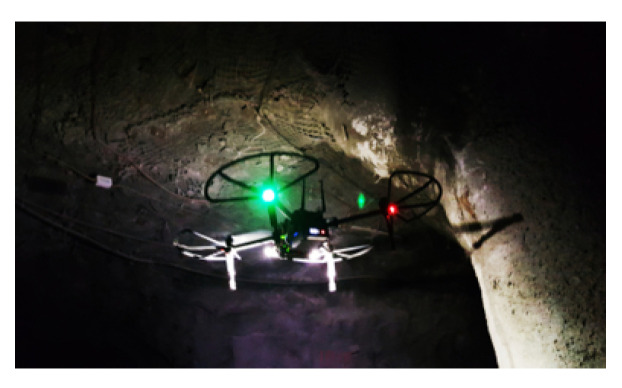
Drone [24].

**Figure 14 sensors-23-04294-f014:**
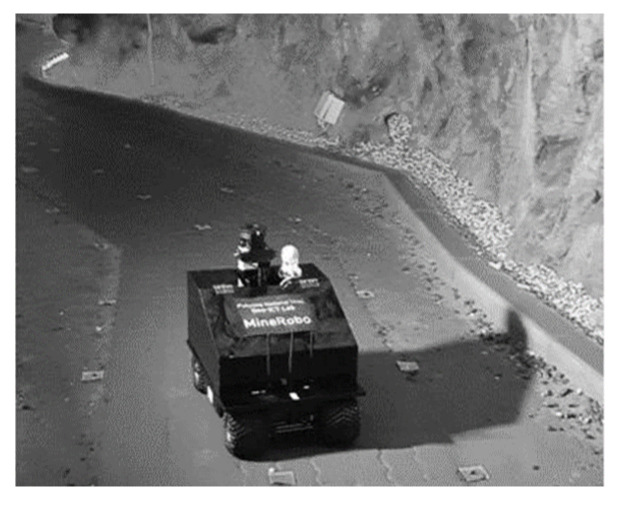
Unmanned ground vehicle [26].

**Figure 15 sensors-23-04294-f015:**
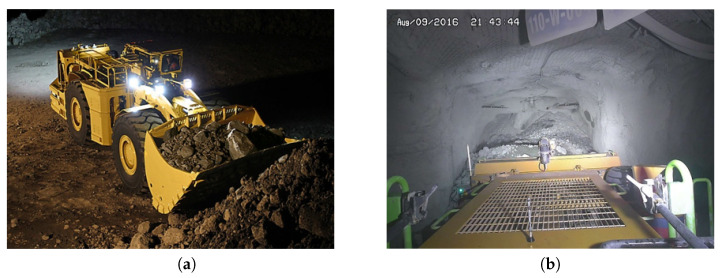
Load–haul–dump [90]: (**a**) load–haul–dump (LHD); (**b**) view from fixed camera on LHD.

**Figure 16 sensors-23-04294-f016:**
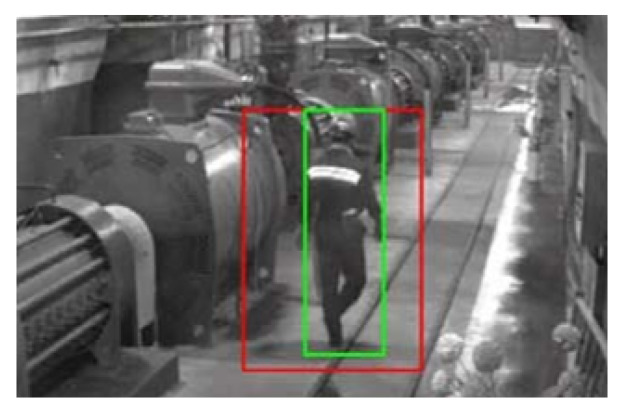
Viewfrom a surveillance camera [95].

**Figure 17 sensors-23-04294-f017:**
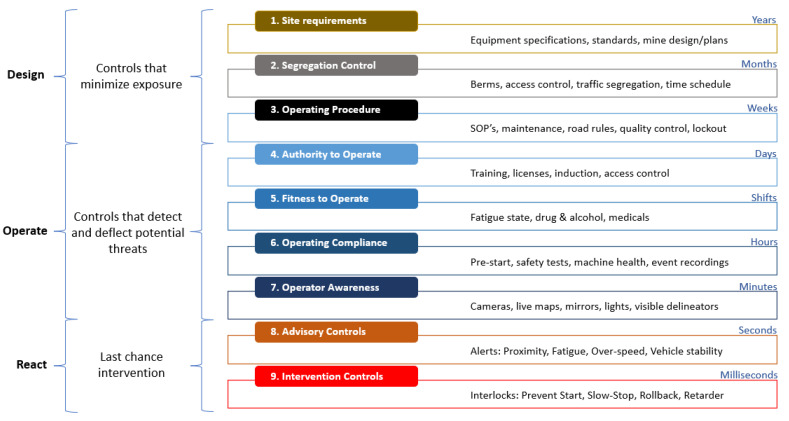
The EMESRT nine layer control effectiveness model reframing our understanding of vehicle interaction controls [120].

**Figure 18 sensors-23-04294-f018:**
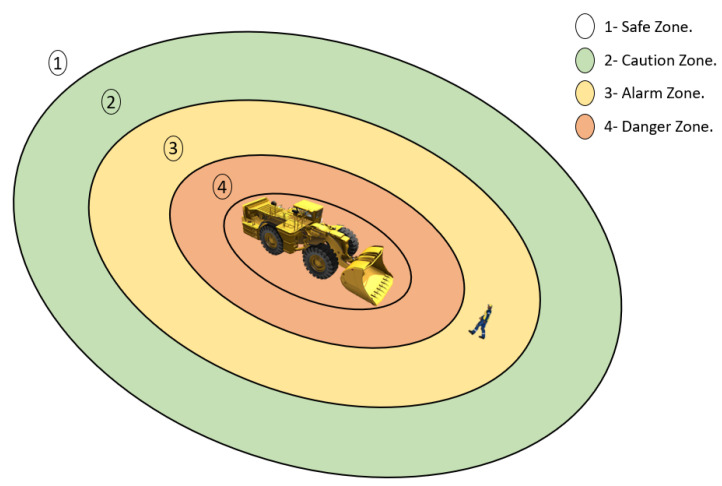
Detection zones around mobile machines during operations.

**Table 1 sensors-23-04294-t001:** Rateof fatal accidents caused by mobile and haulage equipment in some countries.

Country	Period	Fatal Accidents (%)	Causes
USA	1995–2005	31.6%	Haulage Equipment
Australia	1982–1984	29%	Vehicles
United Kingdom	1983–1993	41%	Haulage equipment
Turkey	2004	16%	Haulage equipment
Ghana	2008–2017	48.3%	Mobile Equipment

**Table 2 sensors-23-04294-t002:** Types of accidents by type of machinery.

		Type of Machines	
		Dumper	Load Haul Dump (LHD)
		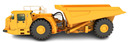	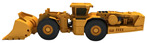
type of accidents	Collision with pedestrian	X	
	Collision with machine	X	X
	(Front/Reversal) Run over	X	
	Fall from machine		X
	Rollover		
	Caught between		X
Country		India	Australia

**Table 3 sensors-23-04294-t003:** Frequency of dumper accidents, according to its mode of action.

Type of Accident	Frequency of Accidents
Reversal run over	51
Front run over	68
Lost control	30
Collision	51
Other	46

**Table 4 sensors-23-04294-t004:** Frequency of load–haul–dump (LHD) accidents, according to its mode of action.

Type of Accident	Frequency of Accidents
Caught between	45
Collision with machine	87
Fall from machine	27

**Table 5 sensors-23-04294-t005:** Metadata of research conducted.

Date	In the initial phase, only papers published between 2000 and 2022 were taken in consideration. The year 2000 was selected after a sensitivity analysis of the quantity of publications identified using the designated keywords.
Paper type	We limited our consideration to research papers.
Language	Our evaluation was restricted exclusively to papers written in English.

**Table 7 sensors-23-04294-t007:** The most commonly used algorithms in underground mines with their purposes.

Algorithm	Data Type	Purpose
Yolo (v2,v3,v4 and v5) [21,22,29,30,31,32]	RGB Images [21,22,30,31,32], Thermal Images [22,29]	Pedestrian detection [21,22,29,30,31,32], electric locomotives and stones falling [30]
HOG [22]	RGB Images [22], Thermal Images [22]	Pedestrian detection [22]
SVM [90,100]	RGB Images [90,100], Thermal Images [100]	Enhancing underground visual place [90], pedestrian segmentation [100]
Image Segmentation and Thresholding [78,100,101]	RGB Images [78], Thermal images [78,100,101]	Overhead boulders detection [78], pedestrian detection [100,101]
Navigation and mapping [23,24,25,26,27,28]	RGB Image [23,25,27], Stereoscopic Image [24,28], Thermal Image [24], Image from LIDAR [23,24,25,26,27,28]	Anti-collision [23], exploration path planning solutions [24], road signs recognition [25], location estimation method [26], trajectory controller [27]

**Table 8 sensors-23-04294-t008:** The most used algorithms with the accuracy they have reached.

Reference	Algorithm	AP (%)	MAP (%)	fps
[21]	Yolov2	N/A	54	43
	YWSSv1	N/A	54.4	43
	YWSS2	N/A	66.3	5
[29]	Yolov4	95.47	N/A	44.3
	SPAD	91.63	N/A	35.9
	Faster-RCNN	88.2	N/A	25.1
	SSD	86.2	N/A	38.7
	Improved Yolov4	69.25	N/A	48.2
[30]	A -Darknet53 YOLOv3	93.1	N/A	31
	B -Model A + add a fourth feature scale	96.1	N/A	26
	C -Model B + Darknet-37	97.5	N/A	38
	D- Model C + DIOU + Focal	98.2	N/A	38

**Table 12 sensors-23-04294-t012:** Summary of industrial collision-avoidance-system-based computer vision.

Company	Product	Type of Solution	Environment	Technology	Comments	Link
DotNetix	SAFEYE	Object detection (recognition signs, pedestrian, mobile machines )	Surface and underground	Stereoscopic cameras, Display warning system, Multicamera controller (mini PC)	dotNetix has created a 3D camera capable of determining the distance to pedestrians (up to 25 m) and machines (up to 50 m).	https://www.dotnetix.co.za/safeye (accessed on 8 March 2023)
	EDEYE	Void and berm detection			EDGEYE provides a berm and void detection system for industrial machines using 3D camera and machine learning.	https://www.dotnetix.co.za/edgeye (accessed on 8 March 2023)
Blaxtair	Blaxtair	Prevents collisions between industrial machinery and pedestrians	Surface and underground	Stereoscopic camera	When a pedestrian is in danger, the Blaxtair Origin delivers a visual and audio warning to warn the driver owing to its AI algorithms, which are supported by a distinctive and large learning database.	https://blaxtair.com/en/products/blaxtair-pedestrian-obstacle-detection-camera (accessed on 8 March 2023)
	Blaxtaire Origin	Pedestrian detection		Monoscopic camera		https://blaxtair.com/en/products/blaxtair-origin-pedestrian-detection-camera (accessed on 8 March 2023)
	Blaxtair Omega	Prevents collisions between industrial machinery and pedestrians		Robust stereoscopic camera (−40 °C to +75 °C)	The Omega 3D industrial camera is a robust tool that computes the disparity map and gives the user access to metadata.	https://blaxtair.com/en/products/omega-smart-3d-vision-for-robust-automation (accessed on 8 March 2023)
AXORA	Radar and video anti-collision system	Collision avoidance	Surface and underground	LIDAR	AXORA provides a LIDAR system that helps vehicles avoid collision in mines using rebounding laser beams to alert operators about obstacles. Additionally, this solution offers a 3D environment map that can be utilized to enhance visibility and exert control over the stability of the ceiling.	https://www.axora.com/marketplace/radar-and-video-anti-collision-system/ (accessed on 8 March 2023)
Sick	Over height and narrowness detection in hard rock shafts	Collision-avoidance system	Underground	2D LIDAR	To avoid collisions with low roofs and in cramped locations, they rely on compact 2D LIDAR sensors to prevent accidents.	https://www.sick.com/cn/en/industries/mining/underground-mining/vehicles-for-mining/over-height-and-narrowness-detection-in-hardrock-shafts/c/p661478 (accessed on 8 March 2023)
	Identification of mine vehicles in areas with poor visibility			LIDAR	The sensors detect when vehicles are dangerously close, even if dust concentrations are high. They also detect whether a vehicle is moving or stationary. They pass the information on to a traffic control system, which warns incoming traffic with a traffic light system if necessary.	https://www.sick.com/cn/en/industries/mining/underground-mining/vehicles-for-mining/identification-of-mine-vehicles-in-areas-with-poor-visibility/c/p674800 (accessed on 8 March 2023)
Torsa	Collision avoidance for shovels	Collision avoidance	Surface and underground	LIDAR 3D	This system is based on LIDAR 3D technology and analyzes the interaction between vehicles and the shovel itself with 0.01 cm of precision, guaranteeing safety in the loading operation by informing the operator of the machinery about the type, position, and distance of the different vehicles and obstacles around the shovel.	https://torsaglobal.com/en/solution/collision-avoidance-shovels/ (accessed on 8 March 2023)
	Collision-avoidance system for haul trucks, auxiliary, and light vehicles				This CAS for trucks and auxiliary vehicles includes LIDAR 3D technology, which is able to analyse its environment with a very high level of precision and definition. The system is designed to protect the vehicle operator at all times, proactively and predictively assessing and alerting potential risk situations.	https://torsaglobal.com/en/solution/collision-avoidance-mining/ (accessed on 8 March 2023)
	Collision avoidance for drillers and rigs				The system has been conceived to identify any object (mining equipment, rocks, personnel, etc.) that could cause a hazardous situation in the driller’s operating area, and it has a powerful communications interface that allows monitoring its safety remotely while operating the rig remotely and autonomously.	https://torsaglobal.com/en/solution/collision-avoidance-drillers/ (accessed on 8 March 2023)
	Collision-avoidance system for underground mining operations			Time of Flight (TOF)	This (CAS) can function over the equipment (Level 9) in underground mining operations to prevent any run-overs or collisions. Covering scoops, auxiliary, and light vehicles, and operational staff are its primary targets.	https://torsaglobal.com/en/solution/collision-avoidance-underground/ (accessed on 8 March 2023)
Wabtec	Collision-management system	Collision-avoidance system	Surface	Camera, RFID, GPS	Wabtec’s Collision Awareness solution is a reporting system developed specifically for the mining industry. In order to reduce the risk of collisions between actors and components of the mine, it gives 360-degree situational awareness of things close to a heavy vehicle throughout stationary, slow-speed, and high-speed operations.	https://www.wabteccorp.com/mining/digital-mine/environmental-health-safety-ehs (accessed on 8 March 2023)
Waytronic Security	Collision avoidance	Collision-avoidance system	Manufacturing	Ultrasonic sensors, camera	Pedestrian and forklift collision avoidance	http://www.wt-safe.com/factorycoll_1.html?device=c&kyw=proximity%20detection%20system (accessed on 8 March 2023)
LSM technologies	RadarEye	Collision-avoidance system	Mining	Radar, camera	Radar sensor with detection range 2–20 m and virtually 360 degree viewing;	https://www.lsm.com.au/item.cfm?category_id=2869&site_id=3 (accessed on 8 March 2023)
Matrix Design Group	IntelliZone	Collision-avoidance system	Underground, coal mine	Magnetic field with optional Lidar/Radar/camera integration	Machine-specific straight-line and angled zones	https://www.matrixteam.com/wp-content/uploads/2018/08/IntelliZone-8_18.pdf (accessed on 8 March 2023)
Preco Electronics	PreView	Collision-avoidance system	Surface and underground mine	Radar, camera	Various product lines	https://preco.com/product-manuals/ (accessed on 8 March 2023)
Caterpillar	MineStar Detect	Collision-avoidance system	Surface and underground mine	Camera, radar, GNSS		https://www.westrac.com.au/en/technology/minestar/minestar-detect (accessed on 8 March 2023)
GE Mining	CAS	Collision-avoidance system	Surface and underground mine	Surface—GPS tracking, RF unit and camera; underground—VLF magnetic and WiFi	Real-time data, data communication network	https://www.ge.com/digital/sites/default/files/downloadassets/GE-Digital-Mine-Collision-Avoidance-System-datasheet.pdf (accessed on 8 March 2023)
Jannatec	SmartView	Collision-avoidance system	Mining	Multi-camera, WiFi Bluetooth (for communication)	Text voice and video communication	https://www.jannatec.com/ensosmartview (accessed on 8 March 2023)
Schauenburg Systems	SCAS surface PDS	Collision-avoidance system	Surface mine	GPS, GSM, RFID, camera	Use time of flight with an accuracy <1 m	http://schauenburg.co.za/product/scas-surface-proximity-detection-system/ (accessed on 8 March 2023)
	SCAS underground PDS		Underground mine	Cameras	Tagless, artificial intelligent	http://schauenburg.co.za/mimacs/ (accessed on 8 March 2023)
Joy Global, P&H	HawkEye camera system	Collision-avoidance system	Mining	Fisheye cameras with infrared filters	Digital video recorder (DVR)—100 to 200 h video	https://mining.komatsu/en-au/technology/proximity-detection/hawkeye-camera-system (accessed on 8 March 2023)
Intec Video Systems	Car Vision	Collision-avoidance system	Industrial	Camera	Vehicle safety monitoring cameras	http://www.intecvideo.com/products.html (accessed on 8 March 2023)
	PreView		Industrial	Radar, camera	Low power 5.8 GHz radar signal	
Ifm Efector	O3M 3D Smart Sensor	Collision-avoidance system	Outdoor	Optical technology	PMD-based 3D imaging information	http://eval.ifm-electronic.com/ifmza/web/mobile-3d-app-02-Kollisionsvorhersage.htm (accessed on 8 March 2023)
Motion Metrics	ShovelMetrics	Collision-avoidance system	Mining and construction	Radar, thermal imaging	Interface with our centralised data analysis platform	https://www.motionmetrics.com/shovel-metrics/ (accessed on 8 March 2023)
3D Laser Mapping	SiteMonitor	Collision-avoidance system	Mining	Laser sensor	Accuracy of 10 mm out of range up to 6000 m	https://www.mining-technology.com/contractors/exploration/3d-laser-mapping/ (accessed on 8 March 2023)
Hitachi Mining	SkyAngle	Collision-avoidance system	Mining	Camera	Bird’s-eye view	https://www.mining.com/web/hitachi-introduces-skyangle-advanced-peripheral-vision-support-system-at-minexpo-international/ (accessed on 8 March 2023)
Guardvant	ProxGuard CAS	Collision-avoidance system	Mining	Radar, camera, and GPS	Light vehicles and heavy equipment	https://www.mining-technology.com/contractors/health-and-safety/guardvant/pressreleases/pressguardvant-proxguard-collision-avoidance/ (accessed on 8 March 2023)
Safety Vision	Vision system	Collision-avoidance system	Diverse range of uses	Camera		http://www.safetyvision.com/products (accessed on 8 March 2023)
ECCO	Vision system	Collision-avoidance system	Diverse range of uses	Camera		https://www.eccoesg.com/us/en/products/camera-systems (accessed on 8 March 2023)
Flir	Vision system	Collision-avoidance system	Wide range of application	Thermal camera		https://www.flir.com.au/applications/camera-cores-components/ (accessed on 8 March 2023)
Nautitech	Vision system	Collision-avoidance system	Harsh environment	Thermal camera	Harsh environment	https://nautitech.com.au/wp-content/uploads/2019/05/Nautitech-Camera-Brochure-2019.pdf (accessed on 8 March 2023)

**Table 13 sensors-23-04294-t013:** Summary of solutions of health monitoring.

Company	Product	Type of Solution	Environment	Technology	Comments	Link
DotNetix	NEXUS	Monitoring operator fatigue	Surface and underground		Using advanced sensors and algorithms, this system monitors the operator and determines if he is distracted or tired from his facial features.	https://www.dotnetix.co.za/fatigue-monitoring (accessed on 8 March 2023)
Torsa	Human vibration exposure monitoring	Miner safety	Surface and underground	Vibration sensor	The TORSA’s vibration monitor system measures and evaluates the vibrations to which the operators of vehicles of the mining operation are exposed for reducing the number of possible injuries that could result from your daily activity.	https://torsaglobal.com/en/solution/human-vibration-exposure-system/ (accessed on 8 March 2023)
Mining3	SmartCap	Monitoring operator fatigue			SmartCap is a wearable device that monitors driver and heavy vehicle operator tiredness. The life app gives operators the ability to control their own level of awareness. Through the user-friendly in-cab life display, the app gives the driver immediate visual and aural alerts.	https://www.mining3.com/solutions/smartcap/ (accessed on 8 March 2023)
Hexagon	MineProtect operator alertness system	Fatigue and distraction management solution	Surface and underground		An integrated fatigue and distraction management solution, the HxGN MineProtect operator alertness system helps operators of heavy and light vehicles maintain the level of attention necessary for long hours and monotonous tasks.	https://hexagon.com/products/hxgn-mineprotect-operator-alertness-system (accessed on 8 March 2023)
Caterpillar	Cat Detect	Fatigue and distraction management	Surface and underground		Any operation is susceptible to distraction and exhaustion, especially if the activities are monotonous. Cat Detect provides fatigue and distraction management, a solution that will help operators create a culture where safety comes first to give you piece of mind.	https://www.westrac.com.au/technology/minestar/minestar-detect (accessed on 8 March 2023)

## Data Availability

Not applicable.

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
