# Peer review of "The Future of Mine Safety: A Comprehensive Review of Anti-Collision Systems Based on Computer Vision in Underground Mines"

_sensors, 2023, doi:10.3390/s23094294_

Round 1

Reviewer 1 Report

- The abstract of a paper should highlight the major need of performing the survey. It should also include the main problems that are going to be discussed in the paper and how the information provided in the paper could benefit a reader.

The abstract should typically include the following elements:

·      Background: A brief statement about the research problem or topic, including the context and significance of the study.

·      Objectives: A clear statement of the main research questions or objectives of the study.

·      Methods: A brief description of the methods or approach used to address the research questions or objectives.

·      Results: A summary of the main findings or outcomes of the study.

·      Conclusions: A statement of the main conclusions or implications of the study.

·      Significance: A brief explanation of the potential impact or benefits of the research findings for the target audience or field of study.

- The introduction of a paper should summarize all the previous techniques used in the discussed field and should contain a major portion of references. However, only a few references have been cited in the Introduction of the paper.

·      A survey paper should not only summarize previous research findings but also critically analyze and evaluate them. Authors should provide their own views and perspectives on the strengths and weaknesses of the technologies or approaches discussed in the survey.

·      In addition, authors should discuss the major issues or challenges identified in previous works and suggest ways to address them in future technologies. This could include identifying gaps in the literature, proposing new research directions or methodologies, or discussing the potential implications of the research findings for the field.

·      Overall, a high-quality survey paper should provide a comprehensive and balanced overview of the state-of-the-art in a particular field, while also providing critical insights and perspectives that can inform future research and development.

-Conclusion should reflect the Abstract of the paper. It should include the major outcomes of the analyses made in the paper, and how this paper is going to provide a good platform for the upcoming strategies in the field.

·       The conclusion should connect the findings and contributions to the broader context of the field, discussing how the paper provides a foundation or platform for future research and development. This could include identifying new research questions or areas of investigation that are suggested by the findings, or discussing how the paper's contributions can inform or improve existing theories, methods, or applications.

·       Overall, the conclusion should provide a concise and compelling summary of the paper's key insights and potential impacts, leaving the reader with a clear understanding of the significance and relevance of the research study. The conclusion should also be closely aligned with the abstract of the paper, reflecting the same key messages and themes that were introduced in the beginning of the paper.

More recent references should be included :

Singh, M., Kumar, M. and Malhotra, J., 2018. Energy efficient cognitive body area network (CBAN) using lookup table and energy harvesting. Journal of Intelligent & Fuzzy Systems, 35(2), pp.1253-1265.

Reviewer 2 Report

The authors propose a survey on anti-collision systems for pedestrian detection in underground mines, and classifies the anti-collision system according to the types of sensors used and their effectiveness in deep underground environment. The main recommendations are as follows: The relevant literature in the mainstream journals in 2022 is not covered enough. It is suggested to supplement the relevant important documents officially published in mainstream journals in 2022 and preprinted.

Reviewer 3 Report

I think the paper is very comprehensive and well-organized. However, I would like to ask the authors to open a new section and propose some new ideas for improving Anti-Collision Systems Based on Computer Vision techniques.

Reviewer 4 Report

The authors present a survey on techniques and the related results for preventing collisions in mines that are based on computer vision approaches. The paper is well written and organized.

Comments:

The first is that the acronyms have to be expanded at their first appearance , for example AP and MAP at lines 434 and 449 respectively.

Second the "one-stage..." please explain the one-stage vs two-stage method, it is not clear now.

Third: a few syntax/grammar errors in lines 37, 244,262, 296, 433, 

Round 2

Reviewer 1 Report

 The authors have implemented sufficient work and the paper could be considered for the journal publication